# *MdMADS6* Recruits Histone Deacetylase *MdHDA19* to Repress the Expression of the Carotenoid Synthesis-Related Gene *MdCCD1* during Fruit Ripening

**DOI:** 10.3390/plants11050668

**Published:** 2022-02-28

**Authors:** Qiqi Li, Ting Wang, Chen Xu, Meishuo Li, Ji Tian, Yi Wang, Xinzhong Zhang, Xuefeng Xu, Zhenhai Han, Ting Wu

**Affiliations:** 1College of Horticulture, China Agricultural University, Beijing 100193, China; liqiqi777@cau.edu.cn (Q.L.); wt001453@cau.edu.cn (T.W.); xuchen@cau.edu.cn (C.X.); s20213172735@cau.edu.cn (M.L.); wangyi@cau.edu.cn (Y.W.); zhangxz@cau.edu.cn (X.Z.); xuefengx@cau.edu.cn (X.X.); rschan@cau.edu.cn (Z.H.); 2Plant Science and Technology College, Beijing University of Agriculture, Beijing 102206, China; 20138501@bua.edu.cn

**Keywords:** MADS transcription factor, protein complex, carotenoid accumulation, fruit ripening

## Abstract

Fruit ripening is regulated by epigenetic modifications and transcription factors, which may function independently or as protein complexes. Our previous study showed that the apple histone deacetylase19 (*MdHDA19*) suppresses fruit ripening through the deacetylation of histones in related genes. Here, a MADS-box transcription factor (*MdMADS6*) was identified using a yeast two-hybrid (Y2H) assay as a candidate protein that interacts with *MdHDA19* during apple fruit ripening. Furthermore, Y2H, bimolecular fluorescence complementation (BiFC) and pull-down assays were used to confirm the interaction between *MdHDA19* and *MdMADS6*. *Agrobacterium*-mediated transient transformation and yeast one-hybrid assays showed that *MdMADS6* promoted carotenoid accumulation in apple fruit by acting on the downstream target genes related to carotenoid biosynthesis. In summary, we conclude that, in the early stages of fruit development, the expression of *MdMADS6* was maintained at lower levels, where it interacted with *MdHDA19* to form a protein complex that inhibited the expression of the downstream genes. At the late stages of fruit development, active expression of *MdMADS6* dissociated the protein complex of *MdMADS6* and *MdHDA19* and consequently promoted the expression of carotenoid biosynthesis genes as well as carotenoid accumulation.

## 1. Introduction

Numerous physiological and biochemical processes occur during apple fruit ripening and affect its quality, post-harvest life and value [1]. Fruits undergo changes in texture, color and aroma during the ripening process. Previous studies revealed that epigenetic modifications, such as DNA methylation, histone acetylation, phosphorylation and ubiquitination, constitute a major factor in regulating fruit ripening [2]. Histone modification is a dynamic process where acetylation is catalyzed by histone acetyltransferases (HATs) and is related to transcriptional activation, whereas deacetylation is catalyzed by histone deacetylases (HDACs) and is involved in transcriptional inhibition [3]. 

Many studies have showed that histone deacetylation plays an important role in regulating fruit ripening. For example, the expression of genes involved in carotenoid and ethylene biosynthesis has been reported to decrease after silencing histone deacetylases1 (*HDA1*) in tomato [4]. The accumulation of carotenoids in fruits of the *histone deacetylases3* (*SlHDA3*)-RNAi lines has been shown to increase simultaneously with up-regulation of the expression of ethylene biosynthesis genes, such as *1-aminocyclopropane-1-carboxylate synthase2* (*ACS2*) and *1-aminocyclopropane-1-carboxylate oxidase1* (*ACO1*) and other genes related to fruit maturation, such as *RIN*, *CNR* and *TAGL1* [5].

In addition to epigenetic modifications, transcription factors, such as members of the MADS-box family as well as the ripening inhibitor (RIN), play an important role in regulating fruit ripening. The sepals of the rin mutant were found to be enlarged, and the fruit ripening was inhibited as a result of the activity of the two MADS-box transcription factors, SlMADS-MC and SlMADS-RIN [6]. Knocking out the *RIN* gene by CRISPR led to synthesis of a kind of RIN-MC protein that inhibited fruit ripening [7]. Another transcription factor, FUL, also plays an important role in fruit development. Overexpression of *PpMADS6*, a homologue of FUL from peach in Arabidopsis, led to early flowering, apical flowers and multi-fruited pods [8]. 

In banana fruits, it was found that *MaMADS1* showed varying expression levels at different stages of fruit development as well as upon harvesting and treatment with exogenous ethylene and 1-methylcyclopropene (1-MCP). Previous studies demonstrated that MaMADS1 is closely related to fruit ripening [9]. Moreover, it was found that the transcription factor SlMADS1 of MADS-Box has a negative regulatory effect on fruit ripening in tomato [10]. In other fruits, such as grapes, strawberries and bilberries [11,12,13], MADS-box transcription factors have been shown to be associated with fruit maturation.

Transcription factors function by activating or inhibiting the downstream genes. *TAGL1* is highly expressed during fruit ripening and early carpel development and can directly bind to the promoter of *SlACS2*, a gene of ethylene biosynthesis, to promote tomato fruit ripening [14]. In citrus, CsMADS6 of the MADS-box family functions as a transcriptional activator. It directly binds to the promoters of the carotenoid biosynthesis-related genes *phytoene synthase* (*PSY*), *phytoene desaturase* (*PDS*) and *carotenoid cleavage dioxygenase1* (*CCD1*) to activate their expression and promote the synthesis of carotenoids. It has also been shown that CsMADS6 has multi-target regulatory effects on carotenoid metabolism [15].

Recent studies have demonstrated that transcription factors can also recruit histone deacetylases to form functional protein complexes. For example, in bananas, the ethylene responsive factor (MaERF11) can initiate MaHDA1 to form an inhibitory protein complex that regulates the expression of maturation-related genes through histone deacetylation, thereby, inhibiting fruit ripening [16]. 

In addition, it was found that the three RPD3/HDA1 subfamily histone deacetylases, SlHDA1, SlHDA3 and SlHDA4, interact with the two MADS-box transcription factors, TAG1 and TM29, in tomato, suggesting that these HDACs may participate in regulating the development and maturation of fruits [17]. In longan (Dimocarpuslongan Lour.) the histone deacetylase (DlHD2) was reported to interact with the ethylene responsive factor (DlERF1), resulting in enhancing the acetylation level of histone H3 during fruit senescence [18].

Our previous data showed that *MdHDA19* inhibits apple fruit ripening. However, the involved regulatory mechanism is still unclear. Therefore, we speculate that *MdHDA19* may interact with transcription factors to regulate apple fruit ripening by acting on downstream target genes.

Here, we identified *MdMADS6*, which can interact with *MdHDA19* and activate the expression of the carotenoid biosynthesis-related gene *MdCCD1* by binding the *CArG* element. Furthermore, we demonstrate that *MdMADS6* and *MdHDA19* may form a protein complex to regulate the ripening of apple fruits.

## 2. Results

### 2.1. MdHDA19 Interacts with MdMADS6

To identify the proteins interacting with *MdHDA19*, we performed a yeast-two-hybrid library screening using *MdHDA19* as bait. A total of 420 interacting proteins were screened (Appendix A), including nine transcription factors. However, according to the protein function and expression levels at various fruit development stages, a MADS-box transcription factor was identified the candidate protein as related to fruit ripening. The MADS-box transcription factor was identified as *MdMADS6* in the Malus domestica through phylogenetic tree analysis and domain analysis (Figure 1A,B).

At the same time, to further investigate the interaction relationship between *MdMADS6* and *MdHDA19*, the vectors pGBKT7-*MdHDA19* and pGADT7-*MdMADS6* were co-transformed into the yeast strain Y2H Gold. The results showed that transformed yeast cells grew on the selective medium lacking Trp, Leu, His and adenine, indicating that *MdHDA19* interacted with *MdMADS6* (Figure 2B). According to the results of a yeast-two hybrid, the *MdMADS6* was verified for further study.

We further tested the direct interaction between *MdMADS6* and *MdHDA19* in vivo by using the bimolecular fluorescence complementation (BiFC) assay. After the fusion proteins of *MdMADS6*-pSPYCE and *MdHDA19*-pSPYNE were co-expressed transiently in tobacco leaf cells, strong YFP fluorescence was observed in the nuclei. As a control, when either of the fusion proteins was co-expressed with the complementary empty vector in tobacco leaf cells, no YFP fluorescence signal was detected, suggesting that *MdMADS6* and *MdHDA19* interacted in tobacco leaves (Figure 2A).

The interaction of *MdMADS6* and *MdHDA19* was studied in vitro using a pull-down assay. The purified His-*MdMADS6* recombinant protein was incubated with GST-*MdHDA19* recombinant protein. The recombinant protein GST-*MdHDA19* was pulled down by His-*MdMADS6* as shown in Figure 2C, indicating that His-*MdMADS6* directly interacted with GST-*MdHDA19* in vitro.

### 2.2. MdMADS6 Promotes Apple Fruit Ripening

To investigate the effect of *MdMADS6* on fruit ripening, we performed the transient transformation assay using *Agrobacterium tumefaciens* to overexpress or silence *MdMADS6* in apple fruits. The fruits with transiently overexpressed *MdMADS6* showed significantly promoted ripening rate (Figure 3A) as well as a higher content of carotenoids (Figure 3C) compared to those transformed with the pRI101 empty vector (Figure 3A). However, the fruits with transiently silenced *MdMADS6* had a suppressed ripening phenotype compared with the control (Figure 3A). In addition, we quantified the expression levels of *MdMADS6* during stages of fruit development. The results revealed that during the early stages of fruit development, the expression levels of *MdMADS6* were significantly lower than those at the late stage (Figure 4A). Therefore, we conclude that *MdMADS6* promotes apple fruit ripening.

### 2.3. MdMADS6 Binds to the Promoter of the Carotenoid Biosynthesis Gene MdCCD1

In previous research, CsMADS6 was found to promote fruit ripening through increasing carotenoid accumulation by targeting the downstream genes involved in carotenoid biosynthesis [15]. In this study, we measured the expression levels of carotenoid biosynthesis genes at different stages of apple fruit development when *MdMADS6* was overexpressed or silenced. The results showed that the expression levels of *MdCCD1* and *MdPDS* were lower at the early than at the late stage of fruit development, especially the expression level of *MdCCD1* was significantly increased at the later stage (Figure 4B). 

In addition, the expression levels of *MdCCD1* showed greater variation in response to modulation of expression of *MdMADS6* than did those of *MdPDS* (Figure 5A). To determine the downstream target genes, we conducted correlation linear analysis between *MdMADS6* and each of *MdCCD1*, *MdPDS* and *MdHYD*. The results showed that the expression levels of *MdCCD1* during fruit development were closest to those of *MdMADS6*. An R^2^ value of 0.7428 (Figure 4C) among the genes *MdPDS* (Figure 4D) and *MdHYD* (Figure 4E), indicated that *MdMADS6* may target *MdCCD1* directly.

To confirm the influence of *MdMADS6* on *MdCCD1* expression, we quantified the expression levels of *MdCCD1* and *MdPDS* in apple fruits in which *MdMADS6* was overexpressed or silenced. The results showed that the expression levels of *MdCCD1* and *MdPDS* increased upon overexpression of *MdMADS6* as compared with the control. However, silencing of *MdMADS6* led to significantly lower expression levels of both genes (Figure 5A), indicating that *MdCCD1* is the possible target gene of *MdMADS6*. We also performed a yeast one-hybrid assay to identify the target gene using pJG4-5-*MdMADS6* and pLaZi-*MdCCD1* (promoter of the *MdCCD1*) co-transformed into the *EGY48* cells. Upon adding the X-Gal staining fluid to the yeast single spots, they turned blue within 6–8 h (Figure 5B), indicating that *MdMADS6* targeting on *MdCCD1* directly. 

Combined with the result of Figure 5A, we concluded that *MdMADS6* targeted *MdCCD1* and positively regulated the expression of *MdCCD1*. To detect the regulating effect of *MdHDA19* on *MdCCD1*, we measured the *MdCCD1* relative expression level in the ‘Orin’ apple calli overexpressed or silenced for *MdHDA19* [19]. The results showed that the expression level of *MdCCD1* decreased compared with the overexpressed pRI101 empty vector but increased significantly compared with overexpressed pRI101-RNAi empty vector, indicating that *MdHDA19* negatively regulated the expression of *MdCCD1* (Figure 5C).

## 3. Discussion

Fruit ripening depends on the regulation of the expression of the involved genes by the concerted action of multiple components, such as transcription factors (activator or repressors) and epigenetic modifiers [20]. Previous studies have shown that histone deacetylation is related to transcriptional inhibition [21]. For example, in tomato, down-regulation of the histone deacetylases SlHDA1 and SlHDA2 promoted ethylene biosynthesis and carotenoid accumulation and thereby accelerated fruit ripening [22]. In our previous research, we found that *MdHDA19* inhibited apple fruit ripening.

To perform their functions, proteins may interact with each other to form functional complexes. Transcription factors may be activated or inhibited by epigenetic modifiers. The *AtHDA5* mutants *hda5-1* and *hda5-2* showed a delayed flowering phenotype, and the expression as well as the acetylation levels of the flowering suppressors FLC and MAF1 were increased. In addition, FVE and FLD interacted with HDA5 and HDA6 indicating that they may form complexes to jointly regulate the flowering time [23].

In the molecular regulatory network of fruit ripening, histone deacetylases could be recruited by the relevant transcription factors to form regulatory protein complexes. For example, in bananas, MaERF11 has been reported to deploy MaHDA1 to regulate ripening-related genes through histone deacetylation. The expression of transcription factor MaERF11 and downstream target genes *MaACO1* and *MaEXP2/7/8* revealed the regulatory mechanism of MaHDA1 in banana fruit ripening [16]. Furthermore, it was found that the three histone deacetylases SlHDA1, SIHDA3 and SlHDA4 interacted with the MADS-box transcription factors TOMATO AGAMOUS1 (TAG1) and TOMATO MADS BOX29 (TM29), which are associated with tomato fruit ripening. 

This suggested that HDACs may be recruited by MADS-box transcription factors to form protein complexes that regulate fruit ripening [24]. In the present study, we screened out the transcription factor *MdMADS6* as interacting with *MdHDA19* through the Y2H assay. We further identified the interaction relationship between *MdMADS6* and *MdHDA19* in vivo and in vitro through Y2H, bimolecular fluorescence complementation and pull-down experiments.

MADS transcription factors have been proven to be important for the regulation of fruit ripening. Among them, RIN and FUL1 are classical members [6,25]. It was found that the transcript levels of *MdMADS6* were low during the early stages of fruit development, whereas it was expressed at higher levels during later stages. The transient transformation experiment proved that *MdMADS6* promoted fruit ripening. Studies have shown that MADS transcription factors can target a variety of downstream genes related to fruit color to regulate fruit ripening, including genes involved in carotenoid biosynthesis [26]. 

In citrus, it was also found that CsMADS6 can activate genes of carotenoid synthesis, leading to carotenoid accumulation, which in turn promoted fruit ripening [15]. In this study, it was found that *MdCCD1*, *MdPDS* and other carotenoid metabolism-related genes were expressed to higher levels at the late stages of fruit development. Moreover, *MdMADS6* interacted with *MdCCD1*, *MdPDS* and other carotenoid metabolism-related genes and positively regulated the expression levels of *MdCCD1* and *MdPDS*. Therefore, we speculated that *MdMADS6* may activate gene expression to promote carotenoid biosynthesis and promote apple fruit ripening.

In conclusion, we propose a two-stage regulatory model of fruit ripening. First, at the early stages of fruit development, the expression of *MdMADS6* is low, which interacts with *MdHDA19* to form an inhibitory protein complex that depresses the expression of the downstream gene *MdCCD1* and, hence, fruit ripening. Second, at the late stages of fruit development, the expression of *MdMADS6* increases, causing the protein complex to dissociate and activates the expression of the downstream genes related to carotenoid biosynthesis to promote carotenoid accumulation and apple fruit ripening (Figure 6).

However, whether the promotion of fruit ripening by *MdMADS6* is related to the acetylation level of *MdMADS6* by *MdHDA19* and how the interaction between *MdHDA19* and *MdMADS6* determines the function of the protein complex have not been addressed. The used assays were not sufficient to specify the stage at which *MdHDA19* and *MdMADS6* dissociated during fruit development. However, our results do confirm the promotion of apple fruit ripening by *MdMADS6*. The data also provide novel evidence for transcription factors that deploy histone deacetylases to regulate fruit ripening.

## 4. Materials and Methods

### 4.1. Plant Materials

*Malus domestica* ‘Gala’ and *MdHDA19*-transgenic ‘Orin’ apple calli were used in this study [19]. Fruits were harvested at 90–110 d after flower blossoming. Fruit peels were sampled and frozen in liquid nitrogen for subsequent analysis. Each sample comprised three biological replicates, and each replicate included three fruits.

### 4.2. DNA Extraction, Total RNA Isolation and cDNA Synthesis

The DNA extraction, total RNA isolation and cDNA synthesis were conducted as previously described [26].

### 4.3. Gene Cloning

The cDNA sequences of *MdHDA19* and *MdMADS6* were downloaded from the Apple Genome Database (https://www.rosaceae.org/, accessed on 15 January 2022). Using gene-specific primers, the coding sequences were cloned from cDNA, and the gene fragments were cloned into TOPO-Blunt vector (Aidlab) and then sequenced (Beijing Shenggong, Beijing, China). The total volume of each PCR reaction was 50 μL, including 25 μL of 2 × Phanta Max Buffer, 1 μL of dNTP Mix, 1 μL of each of the upstream and downstream primers, 1 μL of Phanta Max Super-Fidelity DNA Polymerase, 1 μL of template and ddH_2_O up to 50 μL. The thermal cycler procedure was: 95 °C pre-denaturation for 3 min, 95 °C denaturation for 15 s, 58 °C annealing for 30 s, 72 °C extension for 90 s, and 72 °C final extension for 5 min.

### 4.4. Protein Purification

The pET-32a and pGEX6P-1 vectors were digested with *Ecor*1, and the target gene fragments cloned from ‘Gala’ cDNA were inserted into pET-32a and pGEX6P-1 vectors separately using a seamless cloning method. After confirming the correct sequences, the integrated vectors were transformed into *E. coli* BL21 (DE3) cells. The density of *E. coli* was adjusted to an OD of 0.5. The strong inducer IPTG at a concentration of 0.3 mM was added to induce protein synthesis overnight at 4 °C. Next day, the bacterial cells were broken via sonication and centrifuged, and the supernatant and precipitate were collected. Finally, the supernatant and precipitate were subjected to western blotting. 

After inducing the GST-*MdHDA19* and His-*MdMADS6* protein, the His-tagged protein purification kit (CWBIO, catalog number: CW0894S) was used to pass the purification column to allow GST-*MdHDA19* binding to His-*MdMADS6* and finally perform western blot detection [27]. His-*MdMADS6* was immobilized on the chromatographic column, the solution was passed through the chromatographic column, the flow-through was collected, and the flow-through was incubated with GST antibody. The pull-down assays were performed as previously described [19]. The primers used are listed in Appendix A.

### 4.5. Yeast Two-Hybrid Assay

The yeast library was constructed by Clontech, the yeast two-hybrid vectors used were pGADT7 and pGBKT7, and the used yeast strain was Y2HGold (Shanghai Weidi biotechnology CAT#: YC1002). The vectors pGADT7 and pGBKT7 were digested with Ecor1. The *MdMADS6* cloned fragment was inserted into pGADT7, and the *MdHDA19* cloned fragment was inserted in pGBKT7. The *MdHDA19* gene had no self-activation activity. The constructed baits and prey vectors were introduced into the Y2HGold yeast strain. The yeast cells were spread onto SD/-Leu/-Trp deficient medium and then transferred to the SD/-Leu/-Trp/-His/-Ade medium. The Yeast Two-Hybrid assay was performed as previously described [28].

### 4.6. BiFC Assay

The pSPYCE155 and pSPYNE173 vectors were digested by *BamH*1. The *MdHDA19* cloned gene fragment was inserted into the pSPYNE173 linear vector, and the *MdMADS6* cloned fragment was inserted into the pSPYCE155. Then, the constructed vectors were transformed into *GV3101 Agrobacterium* separately, and the positive clones were picked into 2 mL LB Medium (50 μg/mL Rif, 100 μg/mL Kan). The clones were shaken overnight at 28 °C, transferred into 50 mL of LB medium (50 μg/mL Rif, 100 μg/mL Kan, 10 mM MES and 20 mM acetosyringone) and incubated overnight at 28 °C. Finally, the bacterial cells were collected by centrifugation. The infection buffer (10 mM MgCl_2_, 10 mM MES and 200 mM acetosyringone) was used to re-suspend the solution, the OD600 was adjusted to 1.0, and the cultures were incubated in the dark for 3 h. 

The re-suspended bacterial solutions were mixed in equal proportions and co-injected into tobacco leaves [26]. The leaves were kept for 1 d in the dark and then for 2 d in the light. The leaves were placed around the injection spot on a glass slide and covered with a cover glass. The material was examined under a FV3000 confocal microscope (Olympus) to observe the fluorescence. The excitation and emission wavelengths were 407–457 nm for 4,6-diamidino-2-phenylindole and 514 and 527 nm for enhanced YFP, respectively. The BiFC assay was performed as previously described [27].

### 4.7. Agrobacterium-Mediated Transient Transformation and Treatments

The pTRV (tobacco rattle virus) and the pRI101 vectors were digested with *BamH1* (New England Biolabs, code NO.R3142), and the *MdMADS6* gene cloned fragment was inserted into the overexpression and silencing vectors. The constructed vector was introduced into *GV3101 Agrobacterium*, and positive clones were picked and grown on 2 mL of LB medium (50 μg/mL Rif, 100 μg/mL Kan) with shaking at 28 °C overnight. The cells were then incubated in 50 mL of LB medium (50 μg/mL Kan) Rif, 100 μg/mL Kan, 10 mM MES and 20 mM acetosyringone) at 28 °C overnight and collected by centrifugation. The infection buffer (10 mM MgCl_2_, 10 mM MES and 200 mM acetosyringone) was used to re-suspend the bacterial suspension to an OD600 value of 1.0 and then allowed to stand for 3 h. 

The apple fruits were infected under vacuum (0.08 MPa), eight apples per gene and control were used for each treatment. After 24 h of co-cultivation in the dark, they were cultured under light in a 25 °C constant temperature incubator for about one week to observe the phenotype. After the fruit appeared with a phenotype, pictures were taken, and samples were taken one week later for fluorescence quantitative PCR experiment [26]. In order to facilitate the observation of fruit phenotype, the transient overexpression assay of *MdMADS6* was conducted before the transient silencing *MdMADS6* assay. 

To assess whether increased expression could hasten ripening (color change), fruit were harvested at 90–100 d after flower blossoming for transient overexpression. To assess whether reduced expression would inhibit ripening, fruit were harvested at 100–110 d after the flower blossom stage for transient silencing. The pTRV and pRI101 vectors were preserved in the lab.

### 4.8. Yeast One-Hybrid Assay

The pLacZi and pJG4-5 vectors were digested with *Sal*1 and *Ecor*1, respectively. Then, the *MdCCD1* promoter cloned fragment was inserted into the pLacZi vector, and the cloned fragment of *MdMADS6* was inserted into the pJG4-5 vector. The constructs were co-transformed into EGY48 yeast cells, which were subsequently spread on the -Trp/-Ura two-deficiency medium and incubated for 48 h. After the yeast spots appeared, single spots were transferred to a medium containing the BU component. After the spot grew, 2 μL of X-gal was added [29].

### 4.9. Real-Time Fluorescence Quantification

A reaction mixture consisting of 10 μL of 2 × Taq Pro Universal SYBR qPCR Master Mix (Vazyme, Q712), 0.5 μL of upstream and downstream primers, 1 μL of Template cDNA and finally made up to 20 μL with ddH_2_O was used. Actin (*Malus domestica*) was used as an internal control. The procedure was conducted using ABI QuantStudio™ 6 Flex system (Applied Biosystems Inc., Foster City, CA, USA) with MightyAmp (SYBR Plus) (Takara; code no. R075A). The procedure was as follows: 95 °C pre-denaturation 30 s, 95 °C denaturation 10 s and 60 °C annealing 30 s for 40 cycles, and the melting curve was: 95 °C: 15 s, 60 °C: 60 s and 95 °C: 15 s. Actin was used as an internal reference, and the calculation method of relative expression was 2^−ΔΔCt^, which contained three biological replicates and three technical replicates [30].

## Figures and Tables

**Figure 1 plants-11-00668-f001:**
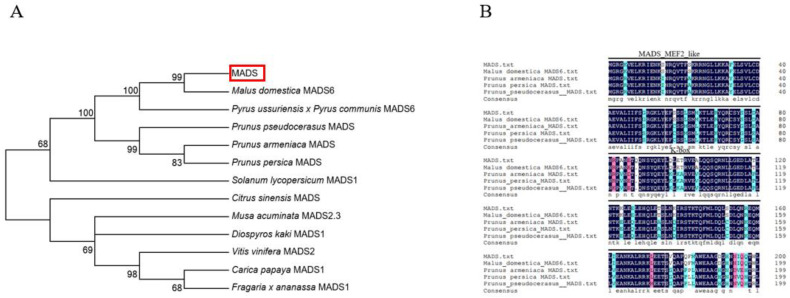
Sequence analysis of the MADS transcription factors. (**A**). Phylogenetic analysis of *MdMADS6* and related proteins from other plant species. The scale bar represents 0.05 substitutions per site. The sequence of MADS was the same as that of MADS6 of the Malus domestica. (**B**). Multiple sequence alignment of *MdMADS6* and related proteins. The horizontal lines mark the two conserved domains.

**Figure 2 plants-11-00668-f002:**
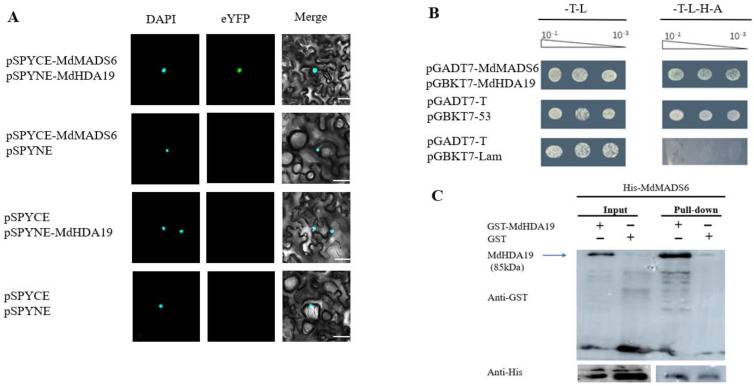
Interaction between *MdMADS6* and *MdHDA19* in vitro and in vivo. (**A**). BiFC assay in tobacco leaf epidermal cells showing the interaction between *MdMADS6* and *MdHDA19* in the living cells. *MdMADS6* fused with pSPYCE and *MdHDA19* fused with the pSPYNE were co-transformed into tobacco leaves and visualized using confocal microscopy. Bars = 10 μm. (**B**). Yeast two-hybrid assay. The coding regions of *MdMADS6* and *MdHDA19* were cloned into the pGADT7 or pGBKT7 vector to create the AD-*MdMADS6*, BD-*MdHDA19*, respectively. The ability of yeast cells to grow on synthetic medium lacking Trp, Leu, His and adenine was scored as a positive interaction. (**C**). Pull-down analysis of the interaction between *MdMADS6* and *MdHDA19* in vitro.

**Figure 3 plants-11-00668-f003:**
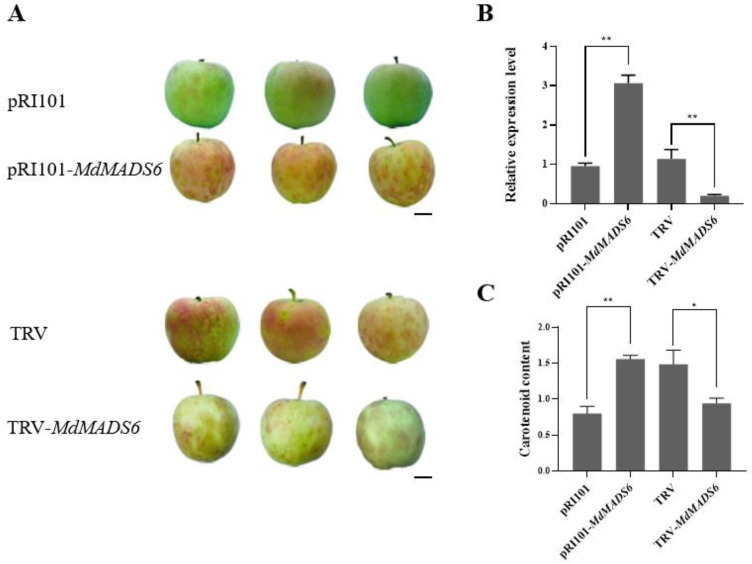
*MdMADS6* promotes apple fruit ripening. (**A**). Phenotypes of ‘Gala’ fruit in which *MdMADS6* was overexpressed (pRI101 vector) or silenced (pTRV vector) by *Agrobacterium tumefaciens*-mediated transient transformation. Bar = 1 cm. (**B**). Gene expression levels of *MdMADS6* after overexpression or silencing in ‘Gala’ apple fruits. (**C**). Carotenoid contents of apple fruits in which *MdMADS6* was overexpressed or silenced by *Agrobacterium tumefaciens*-mediated transient transformation. The transient empty vector was used as a control. Means and SD (*n* = 3) values are shown. * Indicates a statistically significant difference, * *p* < 0.05 and ** *p* < 0.01, as determined by a *t*-test. Error bars indicate SD of three biological replicates and three technical replicates.

**Figure 4 plants-11-00668-f004:**
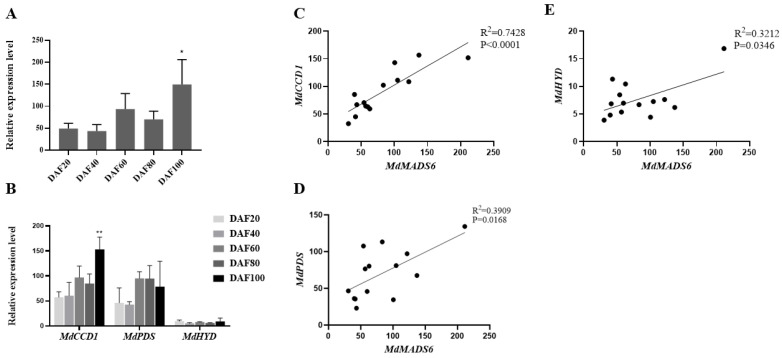
Correlation analysis between *MdMADS6* and relative expression levels of target genes during fruit developmental stages. (**A**). Gene expression levels of *MdMADS6*. (**B**). Gene expression levels of the carotenoid biosynthesis genes *MdCCD1*, *MdPDS* and *MdHYD*. Means and SD (*n* = 3) values are shown. * Indicates a statistically significant difference, * *p* < 0.05 and ** *p* < 0.01, as determined by a *t*-test. Error bars indicate SD of three biological replicates and three technical replicates. (**C**). Correlation analysis of the relative expression levels of *MdMADS6* and *MdCCD1*. (**D**). Correlation analysis of the relative expression levels of *MdMADS6* and *MdPDS*. (**E**). Correlation analysis of the relative expression levels of *MdMADS6* and *MdHYD*. R^2^ means coefficient of determination, *p* value means the significance level, as determined by a *t*-test.

**Figure 5 plants-11-00668-f005:**
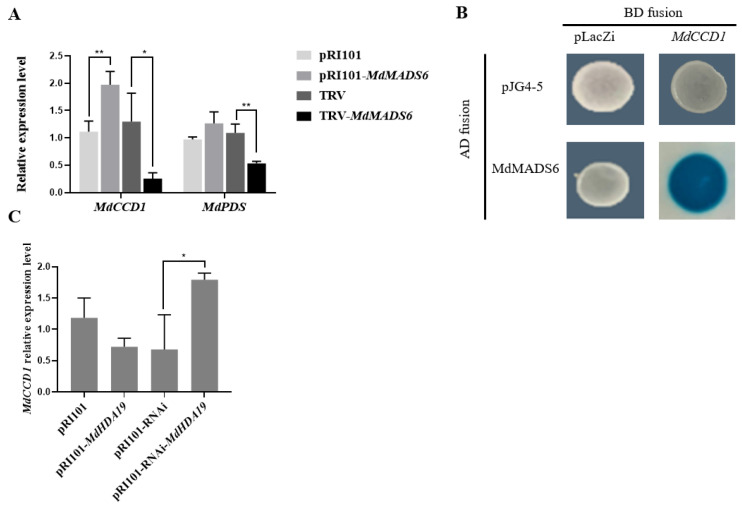
*MdMADS6* binds to the promoter of *MdCCD1* and positively regulates *MdCCD1* expression. (**A**). Gene expression levels of *MdCCD1* and *MdPDS* in the overexpressed or silenced *MdMADS6* ‘Gala’ apple fruits. Means and SD (*n* = 3) values are shown. * Indicates a statistically significant difference, * *p* < 0.05 and ** *p* < 0.01, as determined by a *t*-test. Error bars indicate the SD of three biological replicates. (**B**). Yeast one-hybrid assay. pJG4-5/*MdMADS6* (AD-*MdMADS6*) constructs were co-transformed with pLacZi/MdCCD1 (BD-cis) separately into the yeast cells *EGY48*. AD/BD, AD/BD-cis and AD-*MdMADS6*/BD, were used as negative controls. (**C**). Gene expression levels of *MdCCD1* in overexpressed or silenced *MdHDA19* transgenic ‘Orin’ apple calli. Means and SD (*n* = 3) values are shown. * Indicates a statistically significant difference, * *p* < 0.05, as determined by a *t*-test. Error bars indicate SD of three biological replicates and three technical replicates.

**Figure 6 plants-11-00668-f006:**
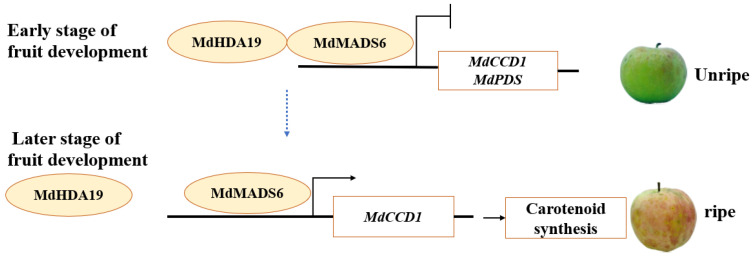
A model showing the influence of a protein complex involving *MdMADS6* and *MdHDA19* on fruit ripening. At the early stages of fruit development, the expression levels of *MdMADS6* are at low levels where it interacts with *MdHDA19* to form a protein complex to inhibit downstream gene expression. At the late stages of fruit development, the active expression of *MdMADS6* dissociates the protein complex of *MdMADS6* and *MdHDA19* activates the expression of carotenoid biosynthesis genes *MdCCD1* and promotes carotenoid accumulation and fruit ripening.

## Data Availability

All data generated or analysed during this study are included in this published article and its Appendix A, which has been added into the article.

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
