# Peer review of "MdMADS6 Recruits Histone Deacetylase MdHDA19 to Repress the Expression of the Carotenoid Synthesis-Related Gene MdCCD1 during Fruit Ripening"

_plants, 2022, doi:10.3390/plants11050668_

Round 1
Reviewer 1 Report
The paper by Li et al. aimed to find targets of MdHDA19 (histone deacetylase) in the regulation of apple fruit ripening.
The authors found a positive interaction of this protein with a MADS-box gene (MdMADS6) in a Yeast-two hybrid screening and further investigated the role of this interaction in genes related to carotenoid biosynthesis.
The paper is well written but misses some important information, such as details about the methods and results.
These are my major concerns:
The data presented in Figure 1 doesn’t add much to the paper. Authors don’t take any results out of it and don’t make any conclusions either. In addition, the sequences in the alignment don’t match with the accession numbers indicated (sequences are full of Ns)
Figure 2C should contain more information to help the identification of the proteins. It misses for example the sizes of the bands
There are no statistical tests in several plots displaying qPCR results (for example 4A). Authors take conclusions that are not supported by statistical analysis. For example, in Figure 4A authors claim that MADS6 expression increases during fruit ripening, but no statistical analysis was performed and there is no information about how many biological replicates were used. In figure 5C authors claimed that the expression level of MdCCD1 decreased in calli overexpressing HDA19, but again this is not supported by the statistical analysis.
The authors should provide more details about the methods used for transient expression in apples. Why are the controls so different in figure 4A (pRI101 and TRV)? I think I know the answer but this should be made clear in the methods at least
Some figures are not referenced in the text. In addition, legends are not complete and should contain more information, to help the comprehension of the results. For example, the statistical tests used, the number of biological replicas, etc…
The authors performed a Yeast-one hybrid experiment showing that MdMADSD6 interacts with MdCCD1 promoter (this information should be added in the figure and legend, because it may seem that authors used MdCCD1 coding sequence, instead of the promoter). Authors claim from this experiment that MdMADS6 could be a positive regulator, yet MdMADSD6 was linked to an activation domain (AD) so I think the author’s claims are incorrect. I guess they can just prove the interaction but not the effect. I think that authors should provide another evidence that this gene is in fact a positive regulator.
Finally, the experiment made in apple calli is incomplete to prove an interaction between MdHDA19 and MdCCD1. Authors should test this using other strategies, for example by transient expression in apples, as they did with MdMADS6.
The quality and resolution of the figures should be improved
Author Response
Q-1: The data presented in Figure 1 doesn’t add much to the paper. Authors don’t take any results out of it and don’t make any conclusions either. In addition, the sequences in the alignment don’t match with the accession numbers indicated (sequences are full of Ns).
R-1: Thank you for your suggestion. Based on the fact that the original image Fig1A and 1B cannot be clearly shown, I replaced the gene bank number in the original image with species Latin and the corresponding transcription factors of the species, at the same time, added and replaced some related horticultural fruit crops.
Q-2: Figure 2C should contain more information to help the identification of the proteins. It misses for example the sizes of the bands.
R-2: We have now added detail information in Figure 2C (now is Figure 1C).
Q-3: There are no statistical tests in several plots displaying qPCR results (for example 4A). Authors take conclusions that are not supported by statistical analysis. For example, in Figure 4A authors claim that MdMADS6 expression increases during fruit ripening, but no statistical analysis was performed and there is no information about how many biological replicates were used. In figure 5C authors claimed that the expression level of MdCCD1 decreased in calli overexpressing MdHDA19, but again this is not supported by the statistical analysis.
R-3: We have now performed statistical analysis for these data. Figure 4A shows the expression level of MdMADS6 in five fruit development stages. Compared with the first stage, the expression level of MdMADS6 was significantly increased in the last fruit development stage. For each fruit development stage, three biological replicates and three technical replicates were used. In the calli overexpressed MdHDA19, the expression level of MdMADS6 decreased, but not significantly through the statistical analysis, by the contrast, the expression level of MdMADS6 increased significantly in the calli with silenced MdHDA19.
Q-4: The authors should provide more details about the methods used for transient expression in apples. Why are the controls so different in figure 4A (pRI101 and TRV)? I think I know the answer but this should be made clear in the methods at least.
R-4: We have now added more information for transient expression in methods session. According to other articles about MADS transcription factor, the gene function of MdMADS6 is predicted to promote ripening in apple fruit. In order to facilitate the observation of apple fruit phenotype, the transient overexpression assay of MdMADS6 was carried out one month before apple ripening to observe whether the ripening was promoted comparing with the pRI101 vector. In the later stage before the ripening, the transient silencing MdMADS6 assay was performed to observe whether it could delay the ripening comparing with the control. The different sample collection times could explain the variation in MdMADS6 expression in TRV and pRI101 control fruit. We have added the detailed description to materials and methods.
Q-5: Some figures are not referenced in the text. In addition, legends are not complete and should contain more information, to help the comprehension of the results. For example, the statistical tests used, the number of biological replicas, etc…
R-5: We have now added related information in the article.
Q-6: The authors performed a Yeast-one hybrid experiment showing that MdMADSD6 interacts with MdCCD1 promoter (this information should be added in the figure and legend, because it may seem that authors used MdCCD1 coding sequence, instead of the promoter). Authors claim from this experiment that MdMADS6 could be a positive regulator, yet MdMADSD6 was linked to an activation domain (AD) so I think the author’s claims are incorrect. I guess they can just prove the interaction but not the effect. I think that authors should provide another evidence that this gene is in fact a positive regulator.
R-6: We have now corrected the information (BD-cis) in the legend to indicate that MdMADS6 interacts with the MdCCD1 promoter, instead of the CDS sequence, in addition, there is a unclear presentation in the article that the result of MdMADS6 positive regulating MdCCD1 is from figure 5A rather than yeast-one hybrid assay.
Q-7: Finally, the experiment made in apple calli is incomplete to prove an interaction between MdHDA19 and MdCCD1. Authors should test this using other strategies, for example by transient expression in apples, as they did with MdMADS6.
R-7: The statement in the article is unclear. We screened the transcription factor MdMADS6 that interacts with it and is related to fruit ripening through MdHDA19, and further screened the downstream target gene MdCCD1 of MdMADS6, and proved that MdMADS6 interacts with MdCCD1 through experiments. We want to further explore whether MdHDA19 had a regulating effect on MdCCD1. Therefore, the expression of MdCCD1 was detected in apple calli that overexpressed and silenced MdHDA19, results showed the negative correlation between MdHDA19 and MdCCD1, so we concluded that MdHDA19 negatively regulates the expression of MdCCD1.
Q-8: The quality and resolution of the figures should be improved
R-8: We have now uploaded the higher quality and resolution image.

Reviewer 2 Report
This manuscript of Li et al. is well-organized, and the results are compelling. The experiment's objective and hypothesis were clear and provided a sufficient amount of data with detailed explanations. The findings are also informative for further studies on fruit ripening. As a result, this manuscript is suitable for publication in Plants.
Author Response
February, 2022
Plants MS ID#: 1583185
MS TITLE: MdMADS6 recruits histone deacetylase MdHDA19 to repress the expression
of the carotenoid synthesis-related gene MdCCD1 during fruit ripening
Dear editor,
Thank you and the reviewers for the constructive comments regarding our manuscript “MdMADS6 recruits histone deacetylase MdHDA19 to repress the expression of the carotenoid synthesis-related gene MdCCD1 during fruit ripening”. Based on the helpful feedback we have now revised our manuscript extensively by incorporating the new results and conclusions. Here, we are resubmitting the revised manuscript to Plants. Below, please find our detailed responses to the questions raised by the reviewers. The major revisions are highlighted in red in the text of the revised manuscript.
Thank you again for your time and effort.
Ting Wu
Reviewer 3 Report
This manuscript describes the identification of a transcription factor that interacts with histone deacetylase (HDA19) in apple to regulate fruit ripening. The manuscript is well written and describes a clear set of experiments potentially providing new insight into the apple ripening process. However, there are several places where additional information would be helpful, information is missing, or is incorrect or incorrectly described.
- line 109. A total of 420 interacting proteins was screened. Given that so many interacting proteins were found why focus on only this one? Please tell the reader in the main text. Were there other transcription factors in this group? Why choose this one? Was it more frequently pulled out?
- Fig 1A. it is difficult to interpret this figure. What are the different proteins shown? From which species? Please provide informative labels or provide more information in the legend. Also, rather than just saying ‘MADS’ on the figure, provide the actual protein name (i.e., MdMADS6).
- Fig 1B. similar to Fig 1A. Provide informative labels for the proteins being compared.
- Fig 2C and lines 131-133. From the data shown it appears that there was ‘pull down’ occurring even in the absence of GST-MdHDA19 raising questions about the in vitro verification of interaction.
- No methods are provided for the pull-down experiments. This should be added.
- Figure 3. What was the developmental stage of the apples used for the overexpression and silencing? This information should be provided both in the results and methods sections. Were samples taken at the same stage (DAF) for both kinds of experiments? If so, why were the controls red for the silencing but green for the over-expressors? How many apples were used for each treatment?
- Fig 3B. At what time relative to inoculation were photographs taken? At what time was expression level determined? Please provide information both in results/figure legend and methods.
- Information should be provided regarding details of the transformation protocol or a protocol reference (i.e. what were the conditions used for the vacuum infiltration). What were the ‘culture conditions’ for the fruit post transformation? Importantly, what was the basis for identifying ‘positive fruits’?
- line 188-192. “The results showed that the expression levels of MdCCD1 and MdPDS increased upon overexpression of MdHDA19 as compared with the control. However, silencing of MdHDA19 led to significantly lower expression levels of both genes (Figure 5A)” the results shown (Fig 5A) are the effect of modulating expression of MADS6 not HDA19. The effect of HDA19 is shown in Fig 5C, but only for CCD1 not for PDS. The legend for Fig 5 should also clearly state which genes are being overexpressed or silenced in each case.
- line 281-283. “Second, at the late stages of fruit development, the expression of MdMADS6 increases, causing the protein complex to dissociate…”. There is no experimental evidence to determine whether there is dissociation of the protein complex, or that it is caused by increased MADS6. It could be that sufficiently increased quantities of MADS6 exceeds the available HDA19 to form a complex, leaving free MADS6. Also, what is known about the levels of HDA19 during fruit ripening? Is this information available from the prior study?
- line 305 states that fruit were harvested at 90 DAF. Please see note 6 above. If all were harvested at the same time, how are the results explained (i.e. some controls red others green)? Also, presumably different ages were used for the developmental analysis and should be clarified in the methods.
- line 315. Specify the primers used for cloning for each of the constructs (reference the list in methods)
- provide references and/or details for protocols for protein purification, yeast two-hybrid, yeast one-hybrid, BiFC, production of transgenic callus. (The failure to provide references does not recognize the work of those who developed them)
- provide source of/reference for TRV vector.
Minor items
- introduction. Sentence line 44 is repetitive with lines 38-39. Suggest changing order to put sentence lines 40-43 before 38-39 and then deleting either 38-39 or 44.
- line 178-179. “The expression levels of MdCCD1 showed greater variation than did those of MdPDS (Figure 5A).” The meaning of this sentence would be clearer if stated: The expression levels of MdCCD1 showed greater variation in response to modulation of expression of MdHDA19 than did those of MdPDS (Figure 5A).
- line 185. ‘…indicated that MdMADS6 targeted MdCCD1’ . this overstates the data. Either ‘suggests that…’ or, ‘indicated that MdMADS6 may target MdCCD1’
- lines 232-239. Not sure what information/value this adds to the discussion, suggest deleting. Also there is a fair bit of repetition in the next paragraph relative to the introduction.
- line 262. A better term would be ‘transient (not instantaneous) transformation’, or ‘transient expression’
- line 279. It is unclear what is meant by ‘MdHDA19 is initiated’ does this refer to the expression of MdHDA19?
Author Response
February, 2022
Plants MS ID#: 1583185
MS TITLE: MdMADS6 recruits histone deacetylase MdHDA19 to repress the expression
of the carotenoid synthesis-related gene MdCCD1 during fruit ripening
Dear editor,
Thank you and the reviewers for the constructive comments regarding our manuscript “MdMADS6 recruits histone deacetylase MdHDA19 to repress the expression of the carotenoid synthesis-related gene MdCCD1 during fruit ripening”. Based on the helpful feedback we have now revised our manuscript extensively by incorporating the new results and conclusions. Here, we are resubmitting the revised manuscript to Plants. Below, please find our detailed responses to the questions raised by the reviewers. The major revisions are highlighted in red in the text of the revised manuscript.
Thank you again for your time and effort.
Ting Wu
Response to Reviewer 3 Comments
Q-1: line 109. A total of 420 interacting proteins was screened. Given that so many interacting proteins were found why focus on only this one? Please tell the reader in the main text. Were there other transcription factors in this group? Why choose this one? Was it more frequently pulled out?
R-1: We identified and sequenced all the yeast spots, then obtained their respective interacting proteins and functional annotations, according to functional annotations, screened out transcription factors related to fruit development, and determinged the candidate proteins using yeast two-hybrid experiments inserted full length of CDS sequence, finally obtained the MADS-box transcription factor, which is function to fruit ripening and interacts with MdHDA19.
Q-2: Fig 1A. it is difficult to interpret this figure. What are the different proteins shown? From which species? Please provide informative labels or provide more information in the legend. Also, rather than just saying ‘MADS’ on the figure, provide the actual protein name (i.e., MdMADS6).
R-2: In order to determine the most homologous protein with the screened MADS, phylogenetic tree analysis was performed in different fruits, and finally it was determined to be MdMADS6, which has been named in apple, but there is no related research. We have now revised related text accordingly.
Q-3: Fig 1B. similar to Fig 1A. Provide informative labels for the proteins being compared.
R-3: The result was re-edited in the article.
Q-4: Fig 2C and lines 131-133. From the data shown it appears that there was ‘pull down’ occurring even in the absence of GST-MdHDA19 raising questions about the in vitro verification of interaction.
R-4: We have now performed additional experiment and replaced the Figure 2C with the new pull down assay.
Q-5: No methods are provided for the pull-down experiments. This should be added.
R-5: We have now added the pull-down detail in methods. In the pull-down experiment, His-MdMADS6 was immobilized on the chromatographic column, the solution was passed through the chromatographic column, the flow-through was collected, and the flow-through was incubated with GST antibody. Western results showed that GST-MdHDA19 protein was detected, but no GST protein was detected, so it could be concluded that MdMADS6 interacted with MdHDA19 in vitro.
Q-6: Figure 3. What was the developmental stage of the apples used for the overexpression and silencing? This information should be provided both in the results and methods sections. Were samples taken at the same stage (DAF) for both kinds of experiments? If so, why were the controls red for the silencing but green for the over-expressors? How many apples were used for each treatment?
R-6: According to the literature, the gene function of MdMADS6 is predicted to promote ripening in fruit. In order to facilitate the observation of fruit phenotype, the transient overexpression assay of MADS6 was carried out one month before apple ripening to observe whether the ripening was promoted compared with the control. In the later stage, the transient silencing MdMADS6 assay was performed to observe whether it could delay the ripening compared with the control. Therefore, the fruit phenotypes of pRI101 and TRV were different, which was determined according to the gene function of MdMADS6. Besides, 8 apples per gene and control were used for each treatment, and the apples with obvious phenotype were picked out to further study. We have added the detailed description to materials and methods.
Q-7: Fig 3B. At what time relative to inoculation were photographs taken? At what time was expression level determined? Please provide information both in results/figure legend and methods.
R-7: We have added the detailed description to materials and methods. In the transient transformation of Agrobacterium, the apple fruit was soaked with the bacterial solution that overexpressed and silenced MdMADS6, and then placed in a constant temperature light incubator for one week after 24 h of co-cultivation in the dark. After the fruit appeared phenotype, pictures were taken, and samples were taken one week later for fluorescence quantitative PCR experiment.
Q-8: Information should be provided regarding details of the transformation protocol or a protocol reference (i.e. what were the conditions used for the vacuum infiltration). What were the ‘culture conditions’ for the fruit post transformation? Importantly, what was the basis for identifying ‘positive fruits’?
R-8: The apple fruits were infected under vacuum, pressure is about 0.08Mpa, and after 24 h of co-cultivation in the dark, they were cultured under light in the 25 ℃ constant temperature incubator for about one week to observe the phenotype. The fruits with obvious phenotypes were selected, and then fluorescent quantitative analysis was carried out, and the positive fruits were screened according to the experimental results.
Q-9: line 188-192. “The results showed that the expression levels of MdCCD1 and MdPDS increased upon overexpression of MdHDA19 as compared with the control. However, silencing of MdHDA19 led to significantly lower expression levels of both genes (Figure 5A)” the results shown (Fig 5A) are the effect of modulating expression of MADS6 not HDA19. The effect of HDA19 is shown in Fig 5C, but only for CCD1 not for PDS. The legend for Fig 5 should also clearly state which genes are being overexpressed or silenced in each case.
R-9: We have now revised the related text.
Q-10: line 281-283. “Second, at the late stages of fruit development, the expression of MdMADS6 increases, causing the protein complex to dissociate…”. There is no experimental evidence to determine whether there is dissociation of the protein complex, or that it is caused by increased MADS6. It could be that sufficiently increased quantities of MADS6 exceeds the available HDA19 to form a complex, leaving free MADS6. Also, what is known about the levels of HDA19 during fruit ripening? Is this information available from the prior study?
R-10: We have now rewriten the related text accordingly.
Q-11: line 305 states that fruit were harvested at 90 DAF. Please see note 6 above. If all were harvested at the same time, how are the results explained (i.e. some controls red others green)? Also, presumably different ages were used for the developmental analysis and should be clarified in the methods.
R-11: The related information about the different time of transiently overexpressed and silenced assays clarified in the method of Agrobacterium-mediated transient transformation and treatments.
Q-12: line 315. Specify the primers used for cloning for each of the constructs (reference the list in methods)
R-12: The cloning primers were listed in the supplemental table 2.
Q-13: provide references and/or details for protocols for protein purification, yeast two-hybrid, yeast one-hybrid, BiFC, production of transgenic callus. (The failure to provide references does not recognize the work of those who developed them)
R-13: The related information and reference have been added.
Q-14: provide source of/reference for TRV vector.
R-14:We have now added the information.
Minor items
- Sentence line 44 is repetitive with lines 38-39. Suggest changing order to put sentence lines 40-43 before 38-39 and then deleting either 38-39 or 44.
R-1:We have now revised the information in the article.
- line 178-179. “The expression levels of MdCCD1 showed greater variation than did those of MdPDS (Figure 5A).” The meaning of this sentence would be clearer if stated: The expression levels of MdCCD1 showed greater variation in response to modulation of expression of MdHDA19 than did those of MdPDS (Figure 5A).
R-2: We have now revised the information in the article.
- line 185. ‘…indicated that MdMADS6 targeted MdCCD1’ . this overstates the data. Either ‘suggests that…’ or, ‘indicated that MdMADS6 may target MdCCD1’
R-3: We have now revised the information in the article.
- lines 232-239. Not sure what information/value this adds to the discussion, suggest deleting. Also there is a fair bit of repetition in the next paragraph relative to the introduction.
R-4: We have now revised the information in the article.
- line 262. A better term would be ‘transient (not instantaneous) transformation’, or ‘transient expression’
R-5: We have now revised the information in the article.
- line 279. It is unclear what is meant by ‘MdHDA19 is initiated’ does this refer to the expression of MdHDA19?
R-6: We have now revised the information in the article. The statement should be clarified that MdMADS6 interacts with MdHDA19 to form a protein complex.

Round 2
Reviewer 3 Report
the revised manuscript addressed most of the concerns. three items still require further clarity:
- (a). what were the other 8 transcription factors pulled down in the yeast two hybrid experiment? Why were they not of interest? In the answer provided in response to the reviewer, it appears that more than one ripening-related transcription factor was pulled out. Why focus on the one that was chosen? (b). For the one protein that was chosen to be of interest based on prior annotation, please provide name of protein of interest and citations for function related to ripening (i.e., your rationale for choosing this one) in the text at the first description of the yeast 2 hybrid experiment. this information should be provided to the reader at the outset, as the choice of that protein was the basis for all of the work that followed.
- The methods and results section still do not clearly indicate age/maturity stage of the apples chosen for the two types of transient assays. Was it based on age (DAF)? If so, provide ages used. Was it based on phenotype of apples (i.e., had obvious signs of ripening prior to the silencing assay)? If so, indicate what signs of ripening were used, e.g., red color development. E.g., To assess whether increased expression could hasten ripening, fruit were harvested at xxx stage. To assess whether reduced expression would inhibit ripening, fruit were harvested at xxx stage.
- The methods state that 8 fruit were treated in each experiment, but only three were used for the data presented. What happened to the other 5 fruit? If only the ‘best fruit’, i.e., those showing greatest phenotype were chosen, this would skew data and cause overestimation of effect. It is understandable that not every infiltrated fruit might have successful transient transformation, however, at a minimum, the number of fruit that did show a phenotype/number of fruit treated should be provided to the reader.
Author Response
February, 2022
Plants MS ID#: 1583185
MS TITLE: MdMADS6 recruits histone deacetylase MdHDA19 to repress the expression
of the carotenoid synthesis-related gene MdCCD1 during fruit ripening
Dear editor,
Thank you and the reviewers for the constructive comments regarding our manuscript “MdMADS6 recruits histone deacetylase MdHDA19 to repress the expression of the carotenoid synthesis-related gene MdCCD1 during fruit ripening”. Based on the helpful feedback we have now revised our manuscript extensively. Here, we are resubmitting the revised manuscript to Plants. Below, please find our detailed responses to the questions raised by the reviewer. The revisions are preserved track changes in the text of the revised manuscript.
Thank you again for your time and effort.
Ting Wu
1.(a). what were the other 8 transcription factors pulled down in the yeast two hybrid experiment? Why were they not of interest? In the answer provided in response to the reviewer, it appears that more than one ripening-related transcription factor was pulled out. Why focus on the one that was chosen? (b). For the one protein that was chosen to be of interest based on prior annotation, please provide name of protein of interest and citations for function related to ripening (i.e., your rationale for choosing this one) in the text at the first description of the yeast 2 hybrid experiment. this information should be provided to the reader at the outset, as the choice of that protein was the basis for all of the work that followed.
Response: We have added the related information in the article.
According to the yeast two-hybrid assay, we screened 420 interacting proteins, including 9 transcription factors, which containing a repetative MADS-box transcription factor and other transcription factors, marked with * in the supplemental table 1. Based on this result, firstly, we identified their gene functions through functional annotations in other horticultural crops and selected candidate proteins related to fruit ripening. Then we verified whether their expression levels had obvious trends in various stages of fruit development through the transcriptome data. At the same time, yeast two-hybrid assay using full length of candidate genes was performed to select transcription factors that interacted with MdHDA19. According to above assays, MADS-box transcription factor was the only one transcription factor for further study.
2.The methods and results section still do not clearly indicate age/maturity stage of the apples chosen for the two types of transient assays. Was it based on age (DAF)? If so, provide ages used. Was it based on phenotype of apples (i.e., had obvious signs of ripening prior to the silencing assay)? If so, indicate what signs of ripening were used, e.g., red color development. E.g., To assess whether increased expression could hasten ripening, fruit were harvested at xxx stage. To assess whether reduced expression would inhibit ripening, fruit were harvested at xxx stage.
Response: We have added the related information in the material and method session according to the reviewer’s suggestion, which now reads: “To assess whether increased expression could hasten ripening (color change), fruit were harvested at 90d-100d after flower blossom for transient overexpression. To assess whether reduced expression would inhibit ripening, fruit were harvested at 100d-110d after flower blossom stage for transient silencing.
3.The methods state that 8 fruit were treated in each experiment, but only three were used for the data presented. What happened to the other 5 fruit? If only the ‘best fruit’, i.e., those showing greatest phenotype were chosen, this would skew data and cause overestimation of effect. It is understandable that not every infiltrated fruit might have successful transient transformation, however, at a minimum, the number of fruit that did show a phenotype/number of fruit treated should be provided to the reader.
Response: Eight apple fruits for each gene were collected for transient transformation assay, and after transformation, they were placed in the dark for one night and in the light for 7 days. Since not every infiltrated fruit might have successful transient transformation, we finally picked out three apple fruits that were transformed successfully according to the phenotype and Real-time fluorescence quantification assay for further study.
